# Sensor Equipped UAS for Non-Contact Bridge Inspections: Field Application

**DOI:** 10.3390/s23010470

**Published:** 2023-01-01

**Authors:** Roya Nasimi, Fernando Moreu, G. Matthew Fricke

**Affiliations:** 1The Department of Civil and Environmental Engineering, University of Nebraska-Lincoln, Lincoln, NE 68583, USA; 2The Department of Civil, Construction, and Environmental Engineering, The University of New Mexico, Albuquerque, NM 87131, USA; 3Department of Computer Science, The University of New Mexico, Albuquerque, NM 87131, USA

**Keywords:** UAS, bridge, computer vision, low-cost sensing, field implementation

## Abstract

In the future, sensors mounted on uncrewed aerial systems (UASs) will play a critical role in increasing both the speed and safety of structural inspections. Environmental and safety concerns make structural inspections and maintenance challenging when conducted using traditional methods, especially for large structures. The methods developed and tested in the laboratory need to be tested in the field on real-size structures to identify their potential for full implementation. This paper presents results from a full-scale field implementation of a novel sensor equipped with UAS to measure non-contact transverse displacement from a pedestrian bridge. To this end, the authors modified and upgraded a low-cost system that previously showed promise in laboratory and small-scale outdoor settings so that it could be tested on an in-service bridge. The upgraded UAS system uses a commodity drone platform, low-cost sensors including a laser range-finder, and a computer vision-based algorithm with the aim of measuring bridge displacements under load indicative of structural problems. The aim of this research is to alleviate the costs and challenges associated with sensor attachment in bridge inspections and deliver the first prototype of a UAS-based non-contact out-of-plane displacement measurement. This work helps to define the capabilities and limitations of the proposed low-cost system in obtaining non-contact transverse displacement in outdoor experiments.

## 1. Introduction

US infrastructure is old and in poor condition, according to the 2021 infrastructure report card published by the American Society of Civil Engineers (ASCE) [1], bridges received a C grade average. Many of the bridges have exceeded their service life. To keep these bridges operating safely, there is an urgent need for technologies that helps managers inspect and assess their bridges safely, rapidly, and cost-effectively [2]. Recently, the ASCE reported that Class II and III railroads invest approximately 25–33% of their revenues into maintenance [3]. Railroad companies are therefore focusing on advances in technology, such as sensors, to increase the safety of their operations and decrease the cost of maintenance. Data collected during inspections help managers make informed decisions, prioritize and allocate their budgets properly, run their operations safely, and increase their profits. Traditional infrastructure inspections mainly depend on human access for visual observation, which depends heavily on experience and training, as well as the frequency of the inspection. 

In the last few decades, private and public entities and owners have introduced the use of sensors in their infrastructure management protocols to overcome the limitations linked to visual inspection. One effective sensor-based method for detecting structural abnormalities is the measurement of dynamic displacement and its change under load [4,5]. However, while sensors assist in collecting objective data in the field, there are challenges associated with deployment, including, but not limited to: difficulties with sensor installation, cost, time requirements, environmental constraints, and power. Traditional displacement measurement methods such as accelerometers and linear variable differential transformers (LVDTs) that require installation on the structure require calibration and are susceptible to environmental effects such as temperature, limited lifespans, and require maintenance and power [6]. Accessing the sensor installation sites can be difficult, time-consuming, and hazardous [7]. Wireless Sensor Networks (WSN), are easier to attach but they still require some degree of access to the structures and have power supply challenges which may require leveraging energy harvesters for optimal performance [8]. Sensors fixed to structures are difficult to reconfigure once in place and are therefore unsuited to changes in measurement design and capabilities, effectively freezing the technology in place.

Non-contact sensing coupled with UASs provide an alternative and have gained popularity in the structural health monitoring of unconventional structures, such as bridges, because they can reach inaccessible structures and collect data safely and quickly and improvements in technology do not require replacing a large number of fixed sensors.

Cameras have become one of the most popular non-contact sensors for bridge inspections, with several researchers using stationary cameras fixed on the ground for bridge inspections [9,10,11,12]. In these studies researchers used stationary cameras to measure the dynamic vertical displacement of the bridges, in their studies, they ignored the out-of-plane motion of the structure and the rotational motion of the camera and they used a fixed scale factor in their methods. Despite their sampling rate and pixel number limitations, cameras are still preferred over other contact sensors because of their light weight, small size, accessibility, and easier data collection. 

On the other hand, the emergence of UASs, along with the attendant accessibility and safety advantages has created new opportunities for enhancing existing monitoring capabilities by incorporating cameras with other sensors into the UAS [13,14,15,16,17]. These studies use images taken by the UAS to assess bridge conditions. UAS integrated cameras have been used for different purposes such as automation of UAS and their automated landing on a moving target using vision based methods [18,19] or to detect cracks and corrosion, produce the Digital Elevation Maps (DEMs) maps of the structure and its soundings, and collect static data generate 3D structural models [20,21,22,23,24,25,26,27,28]. Carrol et al. [20] designed and developed a sensor package that was deployed on a structure by a drone to conduct a vibration assessment of a structure in the laboratory. Most previous work has used red-green-blue (RGB) cameras to identify visually apparent deficiencies in structures or they focus of static behaviors of the bridges rather than their dynamic responses. However, recent studies used camera-equipped UASs to remotely measure vibrations and dynamic displacement of simple structures [29] and wind turbine blades [30]. Most of this initial quantitative research has been limited to controlled laboratory environments.

One of the limited studies focusing on dynamic displacement measurement using a UAS-camera is led by Yoon et al. [31]. They used a low-cost camera on a UAS to conduct an experiment for the measurement of bridge’s vertical dynamic displacement in laboratory, but their methodology assumes that there is not an out-of-plane motion of the structure, and the data are collected in an image plane, therefore cannot measure the out of plane or transverse component of the displacement.

In previous work, we developed a methodology that combines camera and laser data collected by UAS to measure the transverse component of the displacement for a moving structure [32,33,34]. To date, there is no published research using sensor-equipped UAS to measure out-of-plane displacement/transverse displacement on a real-size bridge experiment. This research proposes and tests the first prototype of a sensor-enhanced UAS that is compatible with full-scale experiments to find transverse displacement. These experiments inform the potential and limitations of the system. Our publications detail the progression from laboratory sensor experiments to outdoor trials, culminating in the current paper where we report the results of field trials with a pedestrian bridge in active service. We first developed a laser and camera-based instrument in the laboratory to test the feasibility of measuring dynamic transverse displacement, followed by mounting the instrument on a Mavic Matrice 600 and making modifications to the methodology to accommodate use in a controlled outdoor environment. Outdoor trials were conducted after applying modifications to the UAS including railroad experts’ suggestions and lessons learned from multiple controlled field and laboratory experiments [34]. We named this modified UAS the *new aerial system with intelligent measurement integration II* (NASAMI II). This paper presents new dynamic-displacement results from real-world experiments in an uncontrolled environment using a bridge in service. These field experiments lead in the learning of barriers for implementation, and new design using the field experience in a real bridge. 

The architecture of the paper is organized as follows: Section 2 summarizes the modifications and upgrades applied hardware-wise and software-wise of a UAS-based approach to find dynamic transverse displacement in the true scale bridge considering environmental factors; Section 3 discusses the steps in field experiments including selection and testing validation system and the test bridge; Section 4 presents field data and analysis; Section 5 includes a discussion about system development its limitations and opportunities; and finally, Section 6 summarizes the conclusion of the research.

## 2. Materials and Methods

This paper presents the results of a true scale test with UAS system that is developed to measure dynamic transverse displacement called NASIMI II [34]. The team extended out previous UAS (NASIMI) to operate in a true scale bridge test. According to Canadian National railroad experts, UASs are required to fly 2 m or farther from a railroad bridge. We change the laser head and its data logger to meet the distance requirement. To perform the measurements the UAS pilot is required to maneuver the UAS so that the range-finding laser hits the bridge at a point of interest. Preliminary experiments revealed that it was difficult for the range-finding laser point to be seen under bright lighting conditions. Therefore, we project a highly visible green laser along with the rangefinder to aid in the alignment of the UAS (Figure 1). 

The displacement measuring laser was powered using the 18-V power supply under the UAS, and the data loggers on the UAS were powered with light 9-V batteries. The payload on the UAS, including the weight of the lasers, data loggers, carbon bars, counterweight, and batteries was about 3800 g. The UAS was able to operate for 20 min with the designed payload, while its flight duration without a payload is 30 min. This yields a 19 km flight range for the platform which far exceeds federal aviation administration (FAA) regulations for line-of-sight flights and FCC regulations for maximum transmitter range (5 km). The UAS system in this research is designed to collect dynamic responses of the bridges which are usually short-term missions and can be carried out in minutes. 

To find transverse displacement, laser measurements were corrected using camera readings using the method proposed by Nasimi and Moreu [33]. A checkboard pattern was placed on the ground, tracked by a downward-facing camera, and used as a reference point for the laser. This allowed small UAS movements to be accounted for in calculating the displacement estimation. The UAS pilot attempts to maintain position over the ground target while measuring the distance to the bridge-mounted target. This allows the movement of the UAS (as opposed to the bridge) to be factored out. In this paper because of the increased experiment scale and environmental uncertainty, there were some discontinuities in collected data, which required more data processing. These discontinuities were either because of camera limitations in detection or failure to align the range-finding laser with the target. Laser-target misalignment was caused by drift due to wind and other aerodynamic challenges. We preprocessed images to enhance the detection of the target and discarded misaligned range data. Figure 2 shows the proposed methodology to find transverse displacement using a laser and camera in an uncontrolled real-scale experiment. 

As illustrated in Figure 2, in this research proposed method starts with image contrast adjustment by converting experiment frames to grayscale and then to binary images. This adjustment is made because in large scale experiments video records the data from a far distance and the performance of the detectors decrease. To find transverse data first, the trajectory of the UAS is determined by tracking the checkerboard target on the ground and using the camera model, Equation (1). Camera model helps to convert image pixel information to 3-dimensional world information and is dominantly used to determine the trajectories.
(1)$$s\left[\begin{array}{c} u\\ v\\ 1 \end{array}\right]=P\left[\begin{array}{c} U\\ V\\ Z\\ 1 \end{array}\right]$$
where *s* is the varying scale factor; *u, v* are image points or the pixel coordinates of the checkerboard coordinates in an undistorted frame in pixels; *U, V,* and *Z* are the world or physical coordinates of the camera; and P is camera matrix (Equation (2)), which is calculated using intrinsic and extrinsic of the camera. *U, V, Z* provide the UAV trajectory which is discussed in detail in [31].
(2)$$P=K\left[\begin{array}{cc} R & t \end{array}\right]$$
where *K* is intrinsic to the camera and $\left[\begin{array}{cc} R & t \end{array}\right]$ is extrinsic of the camera for experiment frames and it includes rotational matrix (*R*) and translational vectors (*t*) for each frame. Subsequently, UAV trajectory (translation and rotation) is used to compensate for the movement that the laser undergoes due to UAS movement. To subtract the UAV motion from the laser data, these two sets of data needed to be synchronized and resampled.

## 3. Field Experiments

### 3.1. Selected Validation System

Field implementation of the proposed system required validation. Due to the height of the test bridge, the validation instrument had to be capable of precise measurements from a distance. We selected a high-resolution Laser Doppler Vibrometer (LDV) from Polytec (RSV-150) [35] shown in Figure 3a for ground truth measurements. Several tests were conducted in the laboratory. Figure 3b shows a picture of the experiment conducted for evaluating the ground-based laser.

Before the flight near the bridge, the LDV values were compared with LVDT measurements in the ground. Figure 4a,b show the measurements of LVDT and the LDV collected from a target moving about 10–35 mm at approximately 26 and 55 m, respectively. Though appropriate for field tests, the LDV is comparatively heavy and 5–6 times more expensive when compared to NASIMI II, including the price of the UAS.

### 3.2. Selected Bridge for Field Deployment of Sensor

For field tests, a steel-truss, pedestrian bridge with a concrete deck was selected. The bridge is located at Arroya Del Oso Park, Albuquerque, New Mexico, USA. Permission from the City of Albuquerque to work with a bridge in active use was obtained prior to field testing. The bridge is 6 m tall and 45 m long. Figure 5 shows the bridge and its location.

### 3.3. Understanding the Movements of Selected Bridge

After laboratory evaluation of LDV and finalizing the location of the bridge experiment site, the research team designed a field test to characterize bridge movement using the ground laser. Information collected from the bridge underloading provided a benchmark against which the performance of NASAMI II could be compared. The bridge was subjected to a variety of human loading during this benchmarking, such as harmonic jumps and running by a human subject and bridge displacement was measured using the ground laser. 

The bridge displacement was collected when it was subjected to different loadings such as: running and jumping. The lateral displacement of the bridge was under 1 mm in all cases had the highest amplitude was observed under running, Figure 6 shows the bridge’s lateral displacement when it is subjected to running.

Measured displacement values reached a peak value of 1 mm when subjected to a running load. As the displacement was below the limit of detection by NASIMI II, the additional movement was manually induced in the target. This target could also help to have a wider area for data collection instead of thin bridge piles. 

### 3.4. Experiment

Figure 7 shows a representative field test conducted with NASIMI II. An operator at a non-contact data collection station collected laser and accelerometer data from the UAS during flight using a wireless connection. The UAS collected data while flying between approximately 2 and 4 m from the structure. This was close enough the gather accurate range data but not so close as to endanger the UAS. Wind speeds during the flight were generally calm with wind gusts of 4 m/s. 

## 4. Results

### 4.1. Experiment Data Summary

The team conducted a total of five field experiments. In these experiments, the movement of a manually perturbed target was measured and analyzed. The experiments were conducted on a bridge in service, with the target movement designed to emulate railroad bridge displacements in different amplitudes. The objective of the manual displacement simulations were twofold: (1) the selected target was wider than the steel truss’ vertical cord on the bridge, which allowed easier targeting by the drone; (2) initial bridge tests with LDV showed that the pedestrian bridge displacement was less than a millimeter, which is beyond the capability of the selected low-cost system. Moving the target manually could enable imposing a higher displacement response similar to what is expected in railroad bridges under dynamic loading.

The duration of data collection for each test lasted about 100 s, and the total duration for an experiment including the takeoff and landing lasted about 6–7 min. The batteries needed replacement after experiment three. To better have a comparative study between the time of data collection using the UAS and the time of sensor deployment in the bridge, it is worth noticing that the approximate time for a team of two people in charge of sensor deployment and retrieval at the bridge exceeds the time of flight using the UAS. Furthermore, access to the superstructure at times is not available and climbing gear may be needed. However, there are situations when flying the UAS may not be possible (wind, snow, rain) and this needs to be considered case by case. Table 1 lists the details of the camera measurements collected from 5 experiments conducted in this field experiment.

Processing the data was prioritized based on the amount of collected information during each experiment, including the camera, ground laser and UAS laser. As listed in Table 1, experiment 3 did not result is usable data because the laser data were very noisy. 

### 4.2. Processing of Experiment Frames

Among 5 experiments, two of them were selected for further analysis and checking the proposed system’s performance. Additionally, it is worth mentioning that the video captured from the camera contained a lot of frames that did not have the target inside, therefore those frames were ignored in data processing to save in computation. 

Due to the height of the bridges, we used a large checkerboard for camera motion estimation, however during the frame processing the checkerboard’s size was misdetected in numerous frames. To mitigate this the image contrast was adjusted to have more salient points for detection. Figure 8 shows one of the experiment frames that was adjusted before post-processing. Image adjustment led to better detection performance. we used adjusted frames to find the UAS trajectory. 

We selected a checkerboard target because the down-facing camera in the proposed method captures the ground which is mostly sandy or asphalt and lacks any salient pattern for tracking purposes. 

### 4.3. Summary of Results

This section presents the results of Experiment 2 and Experiment 5, respectively. First, the raw data from the sensors are plotted, including the measurements from the camera, ground reference laser, and UAS-mounted laser. Subsequently, dynamic displacement incidents are detected and some of them were selected for further processing. Finally, the transverse displacement results for each simulated rail car are estimated by the proposed non-contact approach and compared with measurements from the reference ground instrument.

#### 4.3.1. Experiment 2

Figure 9, Figure 10 and Figure 11 show the raw data from camera and sensors for experiment 2 including: UAS translational and rotational motions captured by camera in 3 directions while UAS is hovering over the checkerboard, and raw data collected from the laser on hovering UAS and ground laser. For real-scale experiments, Y direction corresponds to the direction toward bridge, and Z direction is the UAS flight elevation.

Figure 9 shows the estimated 3D position of the UAS with regard to the corner of the checkerboard pattern on the ground in millimeters. The subtraction of the positions of frames from an arbitrary reference frame provides displacement values. The Y axis of the camera represents the direction toward the bridge surface which is in interest for transverse displacement calculations. Additionally, the Z value of the trajectory shows the elevation of the flight which matches the height of the bridge being tested. The higher the UAS flies, the more frames are misdetected due to the increased distance between the checkerboard and UAS.

Figure 10 shows three Euler angles, roll pitch and yaw, calculated using the rotation matrices from the camera’s extrinsic values. The jump in the yaw angle estimation is known as the gimble lock effect which is an aspect of Euler angles [36]. 

Figure 11a,b show the laser measurements from NASIMI II and those from the reference ground instrument, respectively. The red boxes on each signal show the simulated bridge dynamic displacement for each car. Both lasers were measuring the same target (reflective tape) however, not at an identical point because of the hovering motion of the UAS laser. Both lasers have some missed measurements which are marked with red crosses. 

The red crosses or the abnormal jumps in the UAS laser measurements occurred when the laser on the UAS missed the target.

This paper selected the train car displacement events of interest to illustrate the analysis capability of the proposed method in the context of real bridge environments. After obtaining the camera data, signal processing for combining the measurements is conducted as Following the steps shown in Figure 2. Figure 12 shows the proposed system’s estimation of 3 selected bridge displacements compared with an accurate ground laser. The estimations are sorted from smaller amplitudes to larger amplitudes with peak displacements of 31.75 mm, 73.24 mm, and 116.58 mm. These incidents correspond to fourth, second and third car estimations for experiment 2, respectively. The RMS errors are calculated to be 7.015 mm (22.09%), 10.08 mm (13.76%), and 17.65 mm (15.13%) for the fourth, second and third cars, respectively.

Estimates from the low-cost NASIMI II system compared to ground truth values from the reference laser. The peak-to-peak errors were large for some cars compared to the measurements obtained in an outdoor laboratory [30], this was due to the vision-based measurements by the low-cost monocular camera at a greater height above the checkerboard. This higher error in the estimations is caused by an undetected frame in camera pose estimation during video processing. During the analysis, we noticed significant noise in camera pose estimations which lead to the removal of those frames and a consequent decrease in the effective sampling rate of the camera. Figure 13 shows the camera displacement estimation in the bridge direction; the missed frames are marked with red crosses. Video analysis showed that 508 frames out of 2993 frames were undetected for experiment 2.

#### 4.3.2. Experiment 5

Here, we present the result of experiment 5. Similar to the results from experiment 2, raw data from camera and laser measurements are combined to find transverse dynamic displacement, which is not presented for experiment 5. Figure 14 shows the proposed system’s estimation of 3 selected bridge displacements compared to the reference ground laser. The estimates are sorted by amplitude with peak displacements of 26.79 mm, 47.96 mm, and 52.98 mm. These incidents correspond to the first, eleventh and tenth car estimations for experiment 5, respectively. The RMS errors are calculated to be 9.03 mm (33.7%), 9.61 mm (20.03%), and 7.51 mm (14.17%) for the first, eleventh and tenth, respectively.

For experiment 5, estimates from the low-cost system matched ground truth values from the reference laser as well. Figure 15 shows the camera displacement estimation in the bridge direction; the lost frames are marked with red crosses. Video analysis showed that 798 framers out of 3598 frames were left undetected for experiment 5, this can be caused by higher flight elevation above the ground in this experiment. 

## 5. Discussion

This work extended the aerial NASIMI bridge dynamic translation measurement system to incorporate lessons from small-scale outdoor and laboratory experiments and advice from railroad engineering experts. We then evaluated the performance of the new system, NASIMI II, in the field at an actively used pedestrian bridge in comparison to a comparatively costly and cumbersome LDV reference ground-based system. Modifications to NASIMI included replacing the existing laser with higher-range one. Several field trials were conducted of which two were selected for analysis. The UAS measurements corresponded to the reference instrument (Figure 12 and Figure 14). The peak displacement and error values are listed in Table 2. The results showed that the proposed UAS has the potential for non-contact low-cost monitoring of infrastructure, in this case, a bridge. Though the system worked in principle, the field tests highlighted improvements that will be required before the system can be put into production. The experimental limitations of this study include difficulty in aligning the single beam measurement laser with the target while hovering and limited camera resolution. The accuracy of the data from sensors can be better than that of a non-contact approach, and hence the higher costs of sensor deployment can be justified when high-quality data are required, and specific elements of the structure need to be monitored. However, in the case when the bridge access is limited, or not available, the UAS can be a possible alternative under certain circumstances. UAS non-contact displacement is a good option when the global displacement response can be obtained from the mid-span, assuming bridge linearity, and there is no need to collect data from one specific bridge element. In this comparison, UAS data measurement is valuable for global displacement measurements, and sensor data collection is more accurate for specific local displacement measurements. A higher quality camera and optical zoom lens could help to reduce dropped frames when the UAS is approximately more than 6 m above the camera reference point. The use of an April tag as opposed to the checkerboard could also improve relative pose accuracy. Alternatively, pixel intensity-based methods for tracking can be adopted in cases where placing a checkerboard is restricted. An additional issue was that rotor wash caused instability in the UAS when flying close to the ground and structures, making manual piloting challenging. A potential avenue for future work would be to automate camera registration of the ground target. An automated system would be able to hold its position relative to the camera reference and bridge target much more accurately than a human pilot. This research focused on measuring the dynamic response of the structure, but the proposed system can be used for other tasks, such as finding the dynamic properties and input loads of the structures using measured responses. It also has the potential for transfer to others. For example, the UAS system can be used to find the movement of other structures such as wind turbine blades or piles which it is hard for humans to access. Depending on the information of interest, the UAS can be customized by replacing the laser with other sensors such as Light Detection and Ranging (LiDAR), the correction approach with camera motion can still enable the user to find the absolute reading of their sensor. NASIMI II capabilities can also be adapted to different fields related to the safety of critical infrastructure beyond inspection, with developing new capabilities such as Structure for Motion (SfM) or data fusion. Additionally, the UAS system can be equipped to collect information from the construction site and to enable monitoring and quality control and quality assurance in the field. The adaptability of NASIMI II enables the integration of low-cost LiDAR equipment and other data acquisition for the collection of information on difficult-to-access areas in a cost-effective approach.

## 6. Conclusions

This paper presents the results of a bridge test using a UAS that is designed cost-efficiently to find transverse displacement. The low-cost UAS is designed for non-contact inspection of infrastructure. The proposed system integrates the data from a laser and camera mounted on a UAS to find dynamic transverse displacement which is of interest to railroad inspectors but difficult to measure in the field. The proposed UAS, for the first time, demonstrates how a drone-mounted camera laser system can be used to find transverse displacement in a real-scale testbed. The approach corrects the laser’s translation and rotation during measurements and estimates total dynamic transverse displacements, which cannot be collected with the camera or laser alone. A preliminary UAS was constructed and evaluated in a controlled environment. Using lessons from previous experiments and feedback from railroad engineers, the updated version was tested using a bridge in service. The system was redesigned and developed to adopt a higher-range laser more suited for real field conditions and uncertainties. Displacement of a manually moved target that simulated the displacement of railcars is measured and estimates using the proposed UAS system were compared with a relatively costly commercial LDV. The measurements for the two tests were compared and the mismatch between NASIMI II and the LDV varied between 7 and 17 mm. RMS errors were smaller for larger peaks. Due to flight elevation above the ground and the large distance between the camera and checkerboard, there was an increase in undetected frames during experiments. Data analysis showed that the proposed low-cost system can find displacements but when tests are conducted in a true scale and uncontrolled environment there is still room to make progress for a fully implementable system, especially in terms of resolution and missed data due to UAS movement during hovering. This movement was largely attributable to rotor wash and wind. Future work includes increasing camera detection and estimation performance by making use of optical zoom and leveraging measured responses to find structural and load information. Railroad managers are interested in alternative methods to measure displacements, so these results provide a first approach towards advancing and further improving the non-contact reference-free measurement for implementation. 

## Figures and Tables

**Figure 1 sensors-23-00470-f001:**
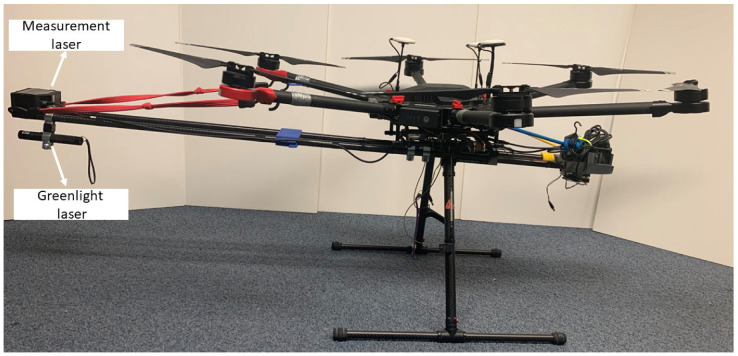
Portable greenlight laser added in the UAS, next to the displacement measuring laser.

**Figure 2 sensors-23-00470-f002:**
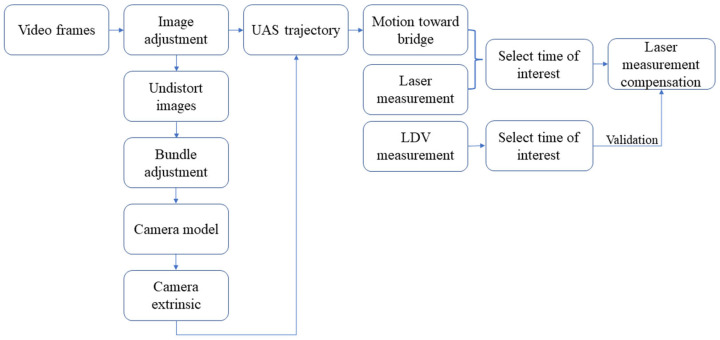
Flow chart of the methodology.

**Figure 3 sensors-23-00470-f003:**
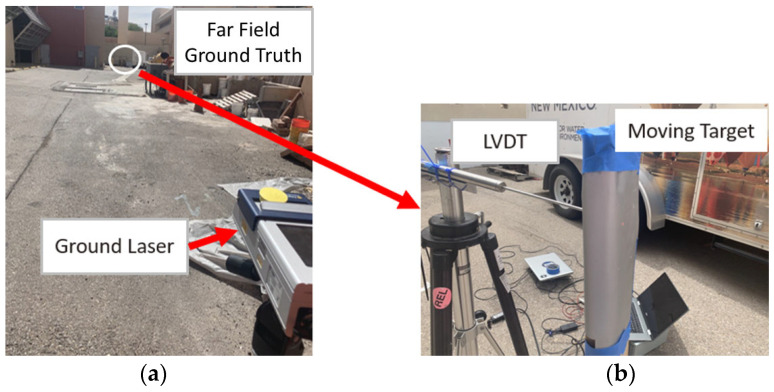
Validation ground laser: (**a**) RSV-150 remote sensing Polytec vibrometer (ground laser) at 55 m; (**b**) Detailed view of the far field ground truth collection with LVDT from moving target.

**Figure 4 sensors-23-00470-f004:**
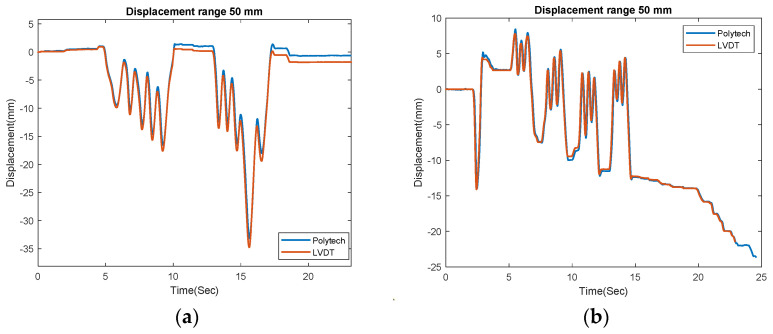
Ground base laser and LVDT measurement of a moving target; (**a**) at 26 m; (**b**) at 55 m.

**Figure 5 sensors-23-00470-f005:**
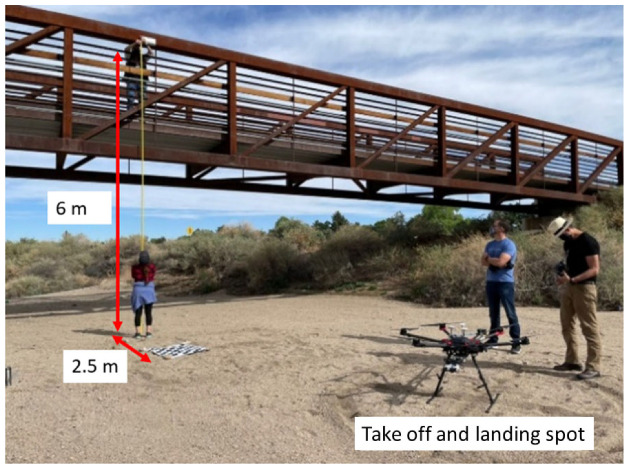
Arroyo Del Oso Park: Test team at the bridge site; NASIMI II in foreground.

**Figure 6 sensors-23-00470-f006:**
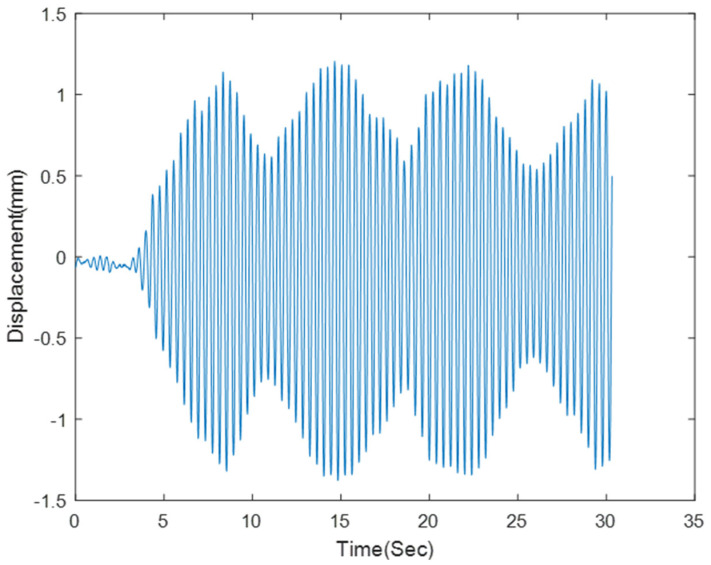
Test bridge’s displacement response collected by Polytec ground laser under: Human running on bridge.

**Figure 7 sensors-23-00470-f007:**
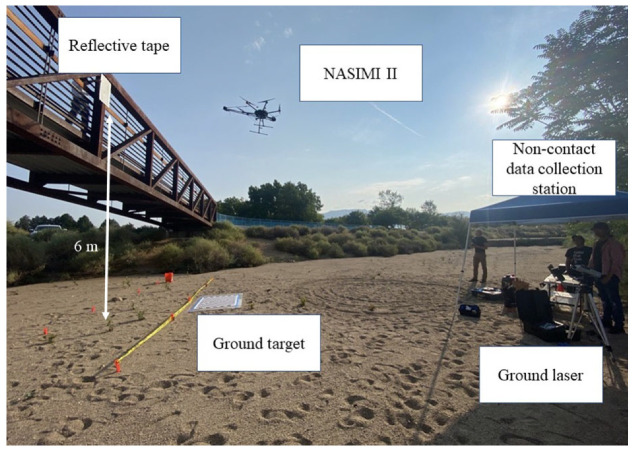
Field deployment of NASIMI II. The DJI Matrice 600 has an airframe diameter of approximately 1.6 m. With the addition of the laser boom the diameter was increased to 2.2 m.

**Figure 8 sensors-23-00470-f008:**
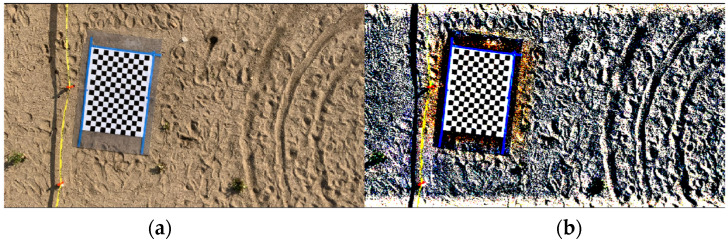
Adjustment of experiment frames before processing them: (**a**) Original frame; (**b**) Adjusted frame.

**Figure 9 sensors-23-00470-f009:**
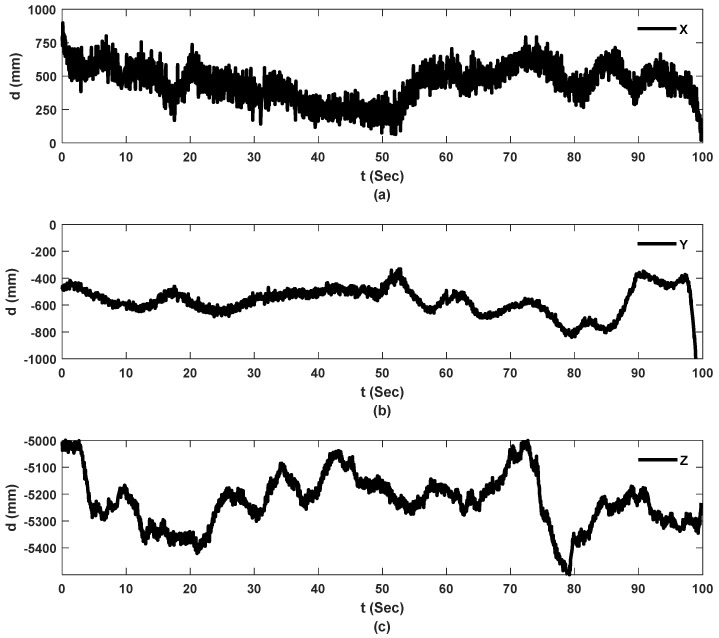
Translational motion of the UAS tracked by camera, experiment 2: (**a**) X direction of the camera; (**b**) Y direction of the camera corresponds to direction of interest (toward bridge); (**c**) Z direction of the camera.

**Figure 10 sensors-23-00470-f010:**
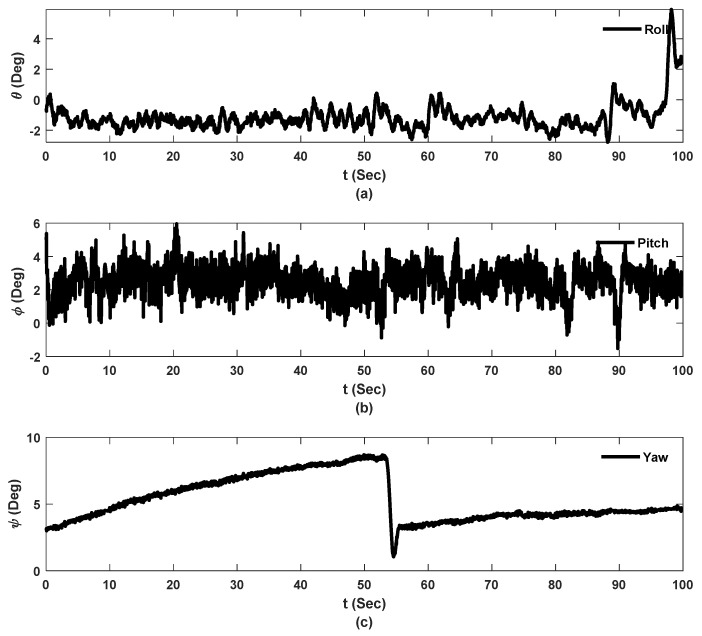
Rotational motion of the system during flight in degrees, experiment 2: (**a**) Roll angle; (**b**) Pitch angle; (**c**) Yaw angle.

**Figure 11 sensors-23-00470-f011:**
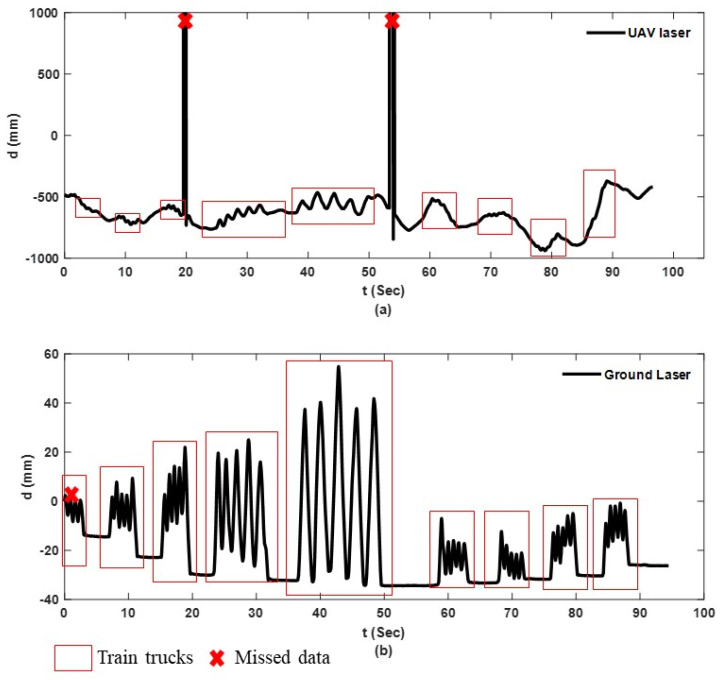
Laser’s raw data, experiment 2: (**a**) Hovering laser on UAS; (**b**) Ground laser.

**Figure 12 sensors-23-00470-f012:**
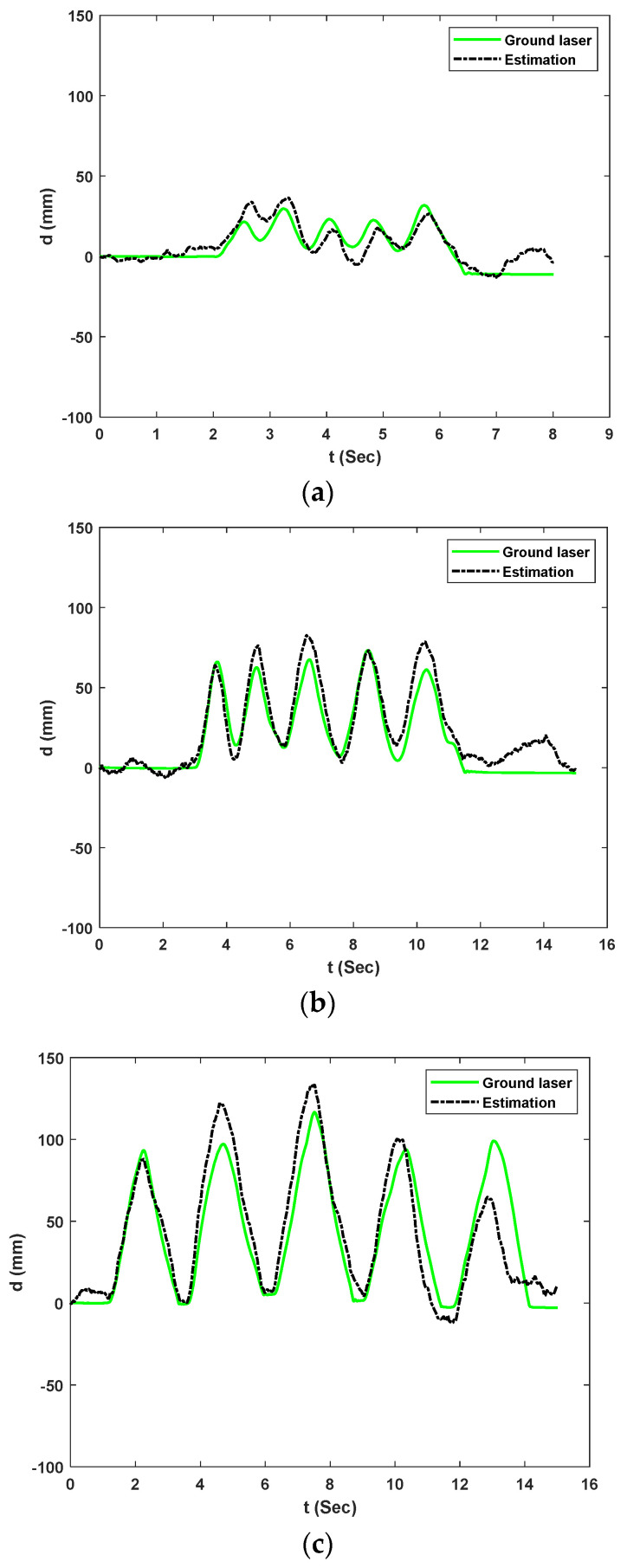
UAS measurement versus the ground laser of the moving board for experiment 2: (**a**) Fourth car displacement estimation; (**b**) Second car displacement estimation; (**c**) Third car displacement estimation.

**Figure 13 sensors-23-00470-f013:**
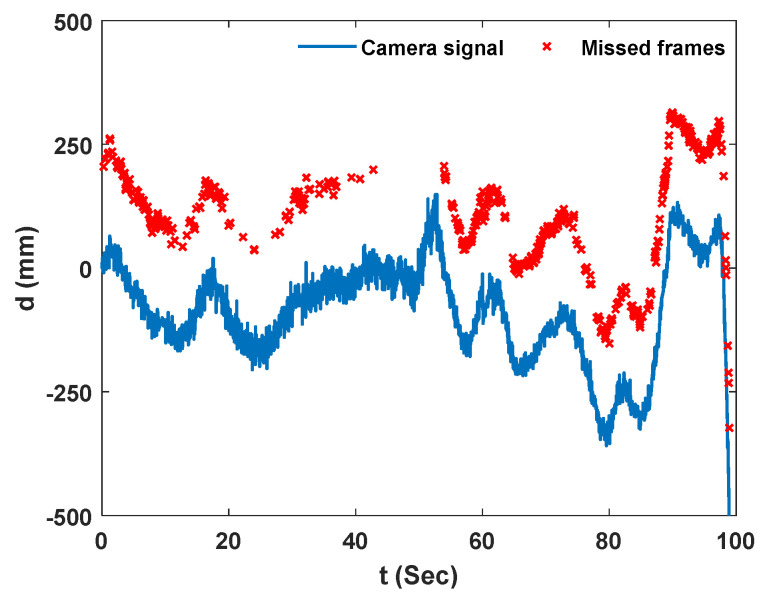
Camera signal in Y direction and the missed experiment frame, experiment 2.

**Figure 14 sensors-23-00470-f014:**
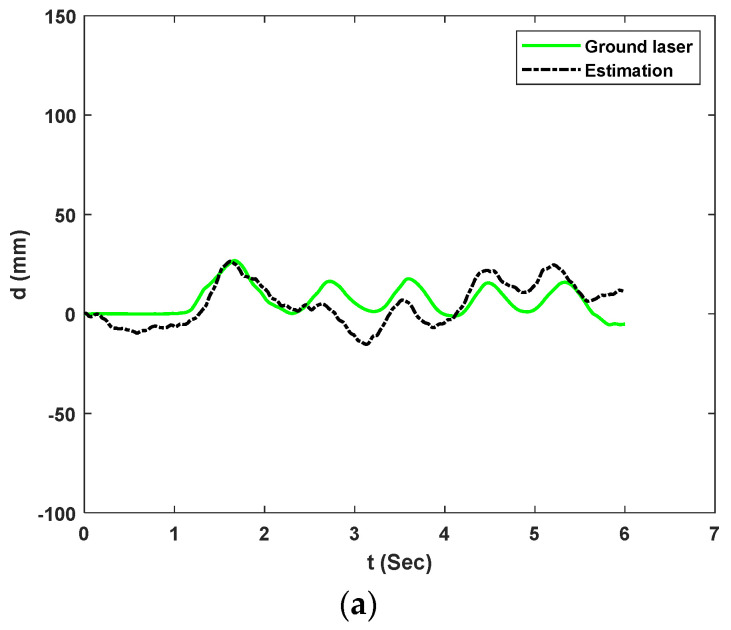
UAS measurement versus the ground laser of the moving board for experiment 5: (**a**) First car displacement estimation; (**b**) Eleventh car displacement estimation; (**c**) Tenth car displacement estimation.

**Figure 15 sensors-23-00470-f015:**
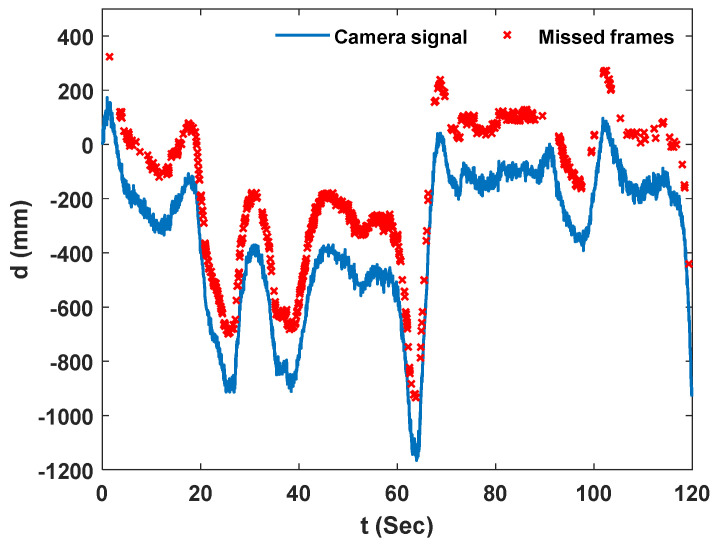
Camera signal in Y direction and the missed experiment frame, experiment 5.

**Table 1 sensors-23-00470-t001:** Summary of the five field experiments, camera data collected, and selection for analysis.

Experiment	Camera Frames	Frames to Analyze	Comments
Test 1	15,799	9113–13,033	Some camera and laser data unavailable
Test 2	16,315	9979–13,194	Camera and laser data aligned
Test 3	10,954	NA (Not Applicable)	Laser data not available for analysis
Test 4	14,951	8175–12,151	Some camera and laser data unavailable
Test 5	9249	3291–7419	Camera and laser data aligned

**Table 2 sensors-23-00470-t002:** Error summaries of experiment.

Experiment 2		Experiment 5	
Peak (mm)	RMS Error (%)	Peak (mm)	RMS Error (%)
31.75	22.09%	26.79	33.70%
73.24	13.76%	47.96	20.03%
116.58	15.13%	52.98	14.17%

## Data Availability

The data presented in this study are available on request from the corresponding author.

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
