# Peer review of "Sensor Equipped UAS for Non-Contact Bridge Inspections: Field Application"

_sensors, 2023, doi:10.3390/s23010470_

Round 1

Reviewer 1 Report

Dear Authors,

please, refer to the attached document. Thanks!

Author Response

The authors thank the reviewers for the specific and very constructive feedback, and we agree that their comments are beneficial to improve the contribution of our work to Sensors journal’s readers. We have addressed all of the comments and believe the updated manuscript is ready for publication. However, we are available for further clarification as needed. Please find attached response document and highlighted manuscript. 

Reviewer 2 Report

In the paper "Sensor equipped UAS for non-contact bridge inspections: field application" the authors undertake an attempt to move a sensor study into the real environment. The first observation is related to the mention of the position in the laboratory, this does not interest anyone and these positions are temporary.

1. The results are strongly dependent on the flight platform used. It is recommended that the methods by which these effects were minimized be presented in detail. This is extremely useful for readers who want to replicate or be inspired by your experiment.

2. Row 166 must be deleted

3. The figures must be arranged from a graphic perspective by arranging them vertically or horizontally as the case may be. For example: Figure 12, 14 can be slightly reduced and all three appear horizontally.

4. I recommend that, in addition to the novelty of the experiments undertaken, any experimental limitations vs. performance in the laboratory by evaluating identical situations.

5. I recommend a comparative study with a situation based on IoT and by placing some sensors on the investigated structure. From this perspective, your application is more futuristic than useful. Anyway, I also encourage such an approach, even if the associated costs currently do not recommend it.

4. The paper is well elaborated and can bring a wave of technological dreaming inspired by cartoons. I think that a comparison with an IoT-based technology from the perspective of costs and the quality of the obtained data is required to correctly define your experiments.

Author Response

The authors thank the reviewers for the specific and very constructive feedback, and we agree that their comments are beneficial to improve the contribution of our work to Sensors journal’s readers. We have addressed all of the comments and believe the updated manuscript is ready for publication. However, we are available for further clarification as needed. Please find attached response document and the highlighted manuscript. 

Reviewer 3 Report

This paper investigates the use of a sensor equipped multicopter UAS to measure non-contact transverse displacement from a pedestrian bridge. Generally, this is an interesting topic. However, from this reviewer’s perspective, it will be better if the authors take the following comments into consideration and revise the paper accordingly.

1.       Any acronym should be defined before use.

2.       In the Introduction, the motive of using a multicopter UAS in the bridge inspection mission should be described in more detail by emphasizing the popularity and superiority of multicopter UASs. The authors may want to refer to some currently published papers by experts in the field, such as Quadcopter precision landing on moving targets via disturbance observer-based controller and autonomous landing planner, Synthesized landing strategy for quadcopter to land precisely on a vertically moving apron.

3.       The contributions of this work, compared to existing related studies, need to be described.

4.       In many real-world cases, it is not easy to place the checkboard right under bridges. What is the solution to such a case?

5.       Is there an explicit mathematical formula for the laser measurement compensation? In order for the paper to convey better to readers, the formula should be added to the manuscript.

6.       There are some typos.  

Author Response

(The authors gave the same response as above.)

Round 2

Reviewer 1 Report

Dear Authors,

Please, refer to the attached document. Thanks!

Reviewer 2 Report

The authors of this paper have made a revised version of much better quality in accordance with the previous recommendations and observations. In this version, the authors prove the scientific quality of the approach as well as its innovative elements.

Reviewer 3 Report

Thanks to the authors for their efforts in revising the manuscript. The updated paper is significantly improved. I have no more comments.